# Perceptions of Arabian Gulf Residents and Citizens about Physical Activity and Social Media Awareness Campaigns: A Qualitative Study

**DOI:** 10.3390/bs14030174

**Published:** 2024-02-23

**Authors:** Ghadah Alkhaldi, Afaf Alotaibi, Rawan Alkasabi, Nourah Alsadhan, Samah Alageel

**Affiliations:** 1Department of Community Health Sciences, College of Applied Medical Sciences, King Saud University, Riyadh 11433, Saudi Arabia; nalsadhan@ksu.edu.sa (N.A.); samalageel@ksu.edu.sa (S.A.); 2Quality Improvement and Patient Safety Department, Prince Sultan Military Medical City, Riyadh 11159, Saudi Arabia; afafalotaibi@psmmc.med.sa; 3Policy Development Department, Council of Health Insurance, Riyadh 11614, Saudi Arabia; ralkasabi@chi.gov.sa

**Keywords:** physical activity, social media awareness campaigns, COM-B model, behavior change, health promotion

## Abstract

Physical activity (PA) is crucial for preventing chronic diseases, but in Gulf Cooperation Council (GCC) countries (Oman, Bahrain, Kuwait, Qatar, Saudi Arabia, and the United Arab Emirates), PA levels are lower than in developed countries. The Gulf Health Council’s social media PA awareness campaign responded to the public’s need for discussion and motivation on this topic. A qualitative study was conducted using semi-structured Zoom interviews with 19 participants from GCC countries between 21 September and 21 October 2021. It aimed to explore PA barriers, facilitators, and perceptions of awareness campaigns. Interviews were transcribed, coded, and analyzed thematically. Facilitators for PA included health value, self-efficacy, persistence, variety, familiar consequences, social support, behavior change techniques, time management, starting at young age, and enjoyment. Barriers encompassed outdoor restrictions, limited amenities, age and weight biases, gym-centric views, lack of proficiency, and injury risk. The study also examined social media awareness campaigns’ effectiveness, identifying themes like engagement, acceptability, reach, design, presentation, and perceived outcomes. Results underscore the complexity of PA facilitators and barriers in the GCC, highlighting the need for campaigns addressing values, perceptions, social connections, and practical challenges, emphasizing the role of research and public policy in boosting PA levels.

## 1. Introduction

The World Health Organization (WHO) defines physical activity (PA) as any movement of the body performed by skeletal muscles that needs energy to happen. This includes activities that meet the definition such as work, household chores, sports, or active play, among others [1]. PA plays a crucial role in maintaining health and well-being. It helps prevent chronic diseases such as obesity, cardiovascular diseases, and diabetes. It also contributes to mental health, reducing symptoms of depression and anxiety. PA improves overall quality of life by enhancing physical fitness, strength, and endurance, and it also offers social benefits, such as increased community engagement and social interaction [2]. 

In the nations that make up the Gulf Cooperation Council (GCC), which are Oman, Bahrain, Kuwait, Qatar, Saudi Arabia, and the United Arab Emirates, the prevalence of overweight and obese individuals, as well as people living with type 2 diabetes, has reached the point where it is now considered an epidemic [3,4,5,6], with the super region where the GCC is located having a prevalence rate of 9.3%, and high body mass index contributes to more than 60% of the type 2 diabetes disability adjusted life years [4].

The levels of PA are likewise quite low compared to the standards set internationally [7]. A review of PA levels in GCC countries found that approximately 39.0–42.1% of men and 26.3–28.4% of women met internationally recommended levels of PA [8], and another review found that physical inactivity exceeded 40% [9].

Barriers to PA in the Gulf region were warm and unsuitable weather, lack of sports facilities or access to them, lack of time, lack of motivation, and certain cultural norms such as those related to gender. Some factors were associated with lower levels of PA including being female, less educated, and elderly [10]. 

Awareness campaigns to promote PA are part of the recommendation by WHO to achieve progress toward active societies and to increase global PA levels [1]. They play a significant role in educating the public about the benefits of PA and the risks associated with sedentary lifestyles. Numerous reviews conducted to explore the effect of campaigns on PA have shown that they have an effect mostly on raising awareness and informing people about the importance of PA [11,12]. They are particularly important in regions like the GCC, where low levels of PA are a concern [7,8,13,14]. These campaigns can utilize various media channels to reach a broad audience, including social media, television, and community events. However, their success depends on their ability to resonate with the target audience, requiring culturally appropriate messaging and an understanding of the local context [12]. Awareness campaigns are most effective when combined with other strategies, such as creating supportive environments and providing accessible facilities for physical activity [2,15]. The element of using social media platforms to deliver the campaign has shown its potential to increase engagement with diverse groups using numerous functionalities, and a recent systematic review showed that social media can be an effective channel for PA interventions [15].

The available research has already identified some barriers and facilitators that affect the GCC community’s participation in PA; some of these were unique to the GCC compared to international studies such as cultural norms and weather [13]. The everyday life processes and perceptions that drive the decision to participate in PA and the perceptions of interventions intended to raise awareness about PA and increase active participation are not yet well understood in that community [13,16]. Qualitative research is invaluable in exploring the complex social, cultural, and personal factors influencing PA in the GCC. It delves deeper than quantitative methods, capturing the nuanced perspectives and lived experiences of individuals [17,18,19]. This research approach is crucial for uncovering specific barriers and motivators for PA unique to the GCC context, such as cultural norms, gender roles, and environmental factors. By understanding these intricacies, health practitioners and policymakers can develop more effective, culturally sensitive interventions to increase PA levels. Hence, this study aimed to identify the PA barriers and facilitators that affect the GCC community and explore community members’ perceptions of PA social media awareness campaigns.

## 2. Materials and Methods

### 2.1. Context

A health awareness campaign was launched by the Gulf Health Council (GHC) under the initiative “For You and for Life” titled “Move for You and for Life” in March 2021. The initiative website address is https://www.ghc.sa/yourhealthguide/awcampaigns/move4youandlife/ (accessed on 22 October 2023). The campaign aimed to promote the health of the GCC community by raising awareness about PA. The main message of the campaign revolved around the idea of simplifying the concept of PA and integrating it as an essential part of individuals’ lives. The campaign’s products were continuously released on various social media platforms until the end of May 2021.

The campaign as a whole targeted adults (aged 20–60 years old) from all segments of the GCC community, and it had the following objectives: spreading the culture of PA in the Gulf community as part of the daily routine, increasing community awareness about different ways of engaging in PA, enhancing community awareness about the various benefits of PA, encouraging institutions to contribute to spreading the culture of PA, empowering individuals to set specific goals related to PA, increasing family support for each other in engaging in PA, and fostering the Gulf community’s interest in PA.

The campaign consisted of an informative video titled ”Your Pulse is Your Guide”, which was aimed at increasing PA, along with various informative messages presented as infographics and animated designs that were shared on social media platforms such as Twitter and Instagram. There was also Voice Coach, which was a podcast dedicated to training walkers and runners for a distance of 5 km or half an hour over nine weeks, a website that included various awareness messages to encourage adherence to a healthy lifestyle and showcasing people’s interactions on social media channels, short informative videos by experts to encourage the adoption of healthy habits, a sports awareness e-booklet that addressed the most important and prominent inquiries about sports, and various mobile wallpapers. 

A mixed methods impact evaluation study was implemented. It included a content analysis of social media posts and a pre- and post-survey for measuring levels of PA and intention to be physically active. The survey was sent to followers of the GHC twitter account. The pre-survey was sent a week before the campaign was launched, and the post-survey was sent in June 2021. The post-survey included an invitation to participate in semi-structured interviews.

### 2.2. Research Design

We conducted a qualitative research study using semi-structured interviews. Data were collected as part of a larger campaign impact evaluation study. Data were analyzed inductively and deductively using thematic analysis.

### 2.3. Study Procedures

As part of the impact evaluation of the campaign, a survey was sent to participants to measure their PA level. The survey invited the respondents to participate in the interviews. To attract a larger number of participants, accounts that commented on the campaign’s tweets from the GHC’s Twitter account were invited to participate. An information sheet and consent form were sent to those who agreed to participate. Once they signed the form, a time and date suitable for the participants was scheduled. Interviews were conducted online (using a secure Zoom meeting). Interviews were audio recorded and transcribed verbatim.

The first interview took place on 8 June 2021, and the last interview was conducted on 21 June 2021. Interview times ranged from 20 to 45 min. Participants filled in a demographic questionnaire to obtain information about their age, sex, country, and occupation. All of the interviews were conducted by Ghadah and Samah (who are Saudi female public health researchers experienced in qualitative research) with assistance from Rawan (female research assistant and health education specialist).

### 2.4. Inclusion Criteria

GCC citizens or residents who completed the survey, or those who commented on campaign’s tweets, were included in the study. 

### 2.5. Instrument

The topic guide was constructed based on the campaign’s evaluation objectives and the COM-B model for behavior change [20]. The model specifies that any behavior to occur requires that the person be capable (C), motivated to perform it (M), and has the opportunity to perform it (O) [20]. The model has been used in numerous studies to explore the reasons behind participation in PA [18,21,22]. The guide consisted of two parts: the first part was about PA, and the second part was about the campaign. The first part explored participants’ perceptions of PA, its definition, and barriers and facilitators to engaging with PA. The questions were based on the COM-B model [20]. The second part consisted of exploring their opinions about the campaign, how to improve upon it, how to disseminate it further (channels to use), personal relevance, usefulness, impact, and recommendations and suggestions (please see Appendix A for the English version of the topic guide). 

### 2.6. Data Analysis 

The researchers transcribed the data verbatim. Four interviews were checked after transcription to ensure transcription accuracy by Ghadah. Then, ATLAS.ti Web was used to manage all transcripts. A mixture of inductive and deductive approaches to coding was used, whereby codes and themes were inductively generated from the data, and then they were analyzed deductively through mapping them to the theoretical framework (COM-B Model) [20]. This approach of starting with an inductive analysis followed by a deductive analysis of data related to the pre-determined framework enables a more rigorous approach to analysis. It reduces the chances of missing some factors not included in the framework or forcing them under constructs that do not fit their meaning. This then allows them to be organized under a framework that enables better interpretation [23,24,25]. 

All the interviews were conducted and analyzed in Arabic. The researchers (Afaf and Ghadah) first became familiar with the data by reading the transcripts and listening to the audio recordings several times to obtain a sense of the substantive content. Data from samples were inductively double-coded; codes were discussed, and changes were made as needed until consensus was reached between coders. The codes were divided into initial themes for classification. Following this, the COM-B model was used as the framework by using it to deductively analyze the data that focused on the barriers and facilitators to physical activity. Ghadah and Afaf reviewed the initial themes and then aligned them with the model’s constructs. Interpretations and thematic frameworks were reviewed and discussed with the rest of the team, who have a background in media campaigns, physical activity, public health, health behavior, and qualitative research. 

### 2.7. Ethical Considerations 

Approval for this research was obtained from the Research Ethics Committee of King Fahd City Medical (21-058E). Participants had to sign the consent form before the interview, and the researcher reiterated the reason for conducting the interviews, along with the participant’s rights, at the beginning of each interview. Throughout the study, the team ensured the confidentiality and anonymity of the participants. Codes, numbers, and dates were used to identify each audio file, and any identifiable data were removed from the transcripts.

## 3. Results

### 3.1. Participant Characteristics 

A total of 76 individuals who provided contact information through the survey were contacted, and 18 of them agreed to participate in the interviews after contacting them with an information sheet and a consent form. Additionally, 27 individuals on Twitter were contacted, and one person accepted the interview invitation. Thus, 19 interviews were conducted with 3 females and 16 males. Among the participants, 42% were from Saudi Arabia. More than half of the participants (63%) were above 30 years old. Table 1 provides a brief background of the participants who were interviewed.

The themes that emerged from the interviews were divided into two parts, PA and campaign. Each theme had subthemes, as shown in Table 2. 

### 3.2. Part 1: Physical Activity 

#### 3.2.1. COM-B Facilitators

##### Health Value of PA 

For some participants, the motivation to be physically active was reflective, as in participants’ PA involved conscious decision-making and setting specific goals to perform it. They thought deeply about the reasons to be physically active and planned for their PA routine. Most participants’ conscious decisions to perform PA involved changing weight, specifically losing it, as one of them said, *“The main stimulus when there is weight gain*.” P10. That was the outcome most participants set as a goal for practicing PA. Few participants’ reasons for performing PA was not only weight per se but general health, as one participant described: *“When I started in 2012, I was 50, now I am 58, to know that with increasing age, activity decreases, and a person begins to lose part of his bones and part of his muscles. This was one of the goals that I tried to preserve the strength of my bones and muscles until I reach a great stage in life. If God gives me longevity.”* P3

##### Self-Efficacy

Self-efficacy, or a person’s belief in their ability to perform PA, played a role in motivating them and in their willingness to overcome barriers in order to be physically active.


*“I envision that, in the first place, there is willpower. This is the most natural thing; when a person has determination and inner conviction, I believe that other matters will be overcome.”*
P4

##### Persistence and Variety

Some participants described how they plan to practice a variety of PA types to maintain their motivation to perform it. One participant explained that the idea of something needs persistence to enjoy its reality.


*“If I love this sport, I find time to do it, I try to enjoy it.”*
P19

##### Familiar Consequences

Sometimes motivation to be physically active was automatic, as it was more instinctive, habitual, or directly linked to immediate outcomes or experiences. Therefore, these participants do not require conscious decision making or explicit goal setting. Such outcomes or effects included self-satisfaction, feeling “*mental comfort*” P1 and/or “*mental clarity*” P8, and improvement in sleep. 

##### Social Support

Having family and/or friends who are physically active is an important social opportunity to sustain and enjoy PA, according to the participants. They also act as a deterrent if the person spends time with those uninterested in PA. Some participants have a norm of participating in group sports when they gather, such as playing football or hiking. For some, it was essential to have a group or an individual that they regularly perform a type of sport or exercise with.


*“You don’t know how to walk alone, meaning you need to have a group. I’m used to always going out with 3–4 people. We go for a walk together… It’s good, but when you’re alone, you become lazy because you have to go by yourself. However, when you’re with a group, you’re more encouraged and your desire to engage in PA or walking or other sports increases.”*
P16

##### Behavior Change Techniques

The person’s capability of performing PA can be psychological. This refers to an individual’s knowledge, skills, and understanding needed to engage in a specific behavior. It involves having the necessary mental and cognitive abilities to facilitate performance of the behavior. Some participants described how they used different behavior change techniques to achieve their desired outcomes, such as setting goals, action planning, weekly monitoring, and self-rewarding.


*“The first step is to set a goal, and I strive to achieve it as much as possible. When the goal is achieved, I reward myself for accomplishing it.”*
P1

##### Time Management 

A skill some participants discussed that is crucial to whether they will be able to be physically active is time management. Although it overlaps with action planning, it was mentioned a lot by participants when they discussed the variety of obligations they have, such as work and family. Being able to decide on how to find time and maintain a schedule for PA is not an easy task. However, it is a cognitive skill to facilitate maintaining PA. 


*“For me, as a married person and an employee, it’s difficult to find free time. However, with some planning, you don’t need to go to the gym every day, and you don’t have to go more than 3–4 times a week. By setting a schedule, it becomes easier to manage your time.”*
P5

##### Starting at a Young Age

Equipping oneself with the skills and knowledge for diverse PA from youth plays a pivotal role in maintaining that momentum as one grows older, as one participant explained, *“During my elementary school days, schools emphasized sports activities, dividing us into teams to play different sports like gymnastics, basketball, volleyball, and football, while competing with other schools. Later, in high school, judo was introduced. These experiences instilled the spirit of sports in us, and we continued to carry that passion forward.”* P14.

##### Enjoyment 

A few of the participants suggested that knowledge about PA is not important as long as it is enjoyable.


*“It is not a condition; for example, in tennis, there are people I see playing who do not have the basics of tennis, but they play and are happy.”*
P19

#### 3.2.2. COM-B Barriers

##### Not a Priority

One barrier for participants was not prioritizing PA compared to other daily activities such as sleeping and praying, so they did not slot it within their daily routine, or that it was not a source of enjoyment for them. Some participants labelled this as procrastination. 


*“Laziness... but no more than that, one would think I’m still going to get up and move, I’m still going to work, I’m still going to engage in physical activity, no—I just sit with my mobile and my laptop—everything is fine and wifi is working that is already enough.”*
P13

When participants were asked whether not having the skills or knowledge to be physically active was an obstacle or barrier, a few thought that there are resources online to learn from and coaches to train with, and that not being physically active is due to there being no motivation to push them. 


*“Well, I mean, we know how to use YouTube now. You just enter and type exercises for the abdomen, exercises for the legs—anything you want will come up—and you’ll see how to do them—we know everything… Nowadays, the world is open, any information you want to find on the internet, you’ll find it. But the important thing is the application and practice, and the application and practice are not there because there’s no motivation—unfortunately, there’s no motivation—and yet, we know the things that cause diseases, we know…”*
P12

##### Outdoor Restrictions 

Physical opportunities are external factors and conditions related to the environment and resources that influence an individual’s ability to engage in PA, specifically walking, exercise, and/or sport participation. Physical opportunities had many barriers that deterred participants from practicing. The barriers depended on the type of outdoor environment; if it was an outside one, such as parks or walkways, which were not available depending on the location. For example, rural areas did not have large parks or pedestrian walkways in which to practice different activities, such as riding bicycles. Even in urban areas, sometimes there were no available spaces that were not polluted by noise from traffic and by car fumes. Walkways were not always designed to accommodate pedestrians, although this is slightly changing, as one participant explained.


*“For example, in (name) Street, the sidewalks are spacious for walking, but the noise there makes it uncomfortable to walk. So, it’s better to have some efforts in terms of permits and new roads, where priority is not only given to cars, making the streets a bit more than just highways. I remember there were narrow streets that have now become highways, but they should leave space for people to walk. …but now there are large sidewalks and it is the beginning, it’ll take time let’s see…”*
P19

In addition to the structure of the outdoor environment, some places are not suitable for different genders and age groups due to cultural and religious reasons.


*“There are no designated places for walking for the elderly, especially knowing that walking for older people requires suitable spaces, as you know.”*
P3


*“Plan to create a pathway, a specific distance, with proper lighting… This pathway is primarily designed for women, so they can walk in a well-lit and safe environment, away from men.”*
P16

Weather is an important factor in deciding to perform PA outdoors. In cooler seasons, people are more likely to be physically active, while in warmer seasons, they can still be physically active, but only during specific times such as very early in the morning. 


*“Weather is an obstacle, but you can overcome it if you go out at dawn, the weather then is nicer.”*
P1

##### Limited Amenities

Indoor places include fitness centers (gyms) or houses. Participants mentioned a lot of conditions that are needed in order to make a fitness center a suitable place for training, such as its geographical location, not being very crowded, being affordable, having different and varied equipment, and being suitable for different members of the family. One participant noted that the latter option was not available everywhere, but it was an element that made it very unique.


*“I entered a gym in (name of city) and saw the facilities there. …they were motivational facilities. For example, a friend of mine and his son are members of a club, so when he takes his son there, they have something for the father, like a special place for him to train while his son is training. So, both the father and the son train together, which is something difficult to find in other clubs in the Gulf countries. And, in this club, the membership fee is paid by the father, and the son can train for free. It’s amazing.”*
P15

##### Age and Gender Bias

Society has certain expectations and perceptions about PA, and some of these might act as a barrier, such as society’s perception on aging. the perception is that the older the person is, the less likely they should be physically active. Hence, opportunities for younger people are plentiful compared to those for the elderly. However, this is currently changing depending on the country. 


*“As for sports clubs, they now only take young people aged 17, 18, and those in their twenties. They don’t take anyone older… their focus is on winning championships, but they should instead create commercial swimming pools or regular commercial playgrounds. There’s no problem in utilizing them to generate income and motivate people to engage in sports. Clubs haven’t really motivated people to participate in sports; it’s all about the finals.”*
P15


*“On the contrary, as you grow older, you may need to exercise even more. In places like Europe or other countries, sports are not limited to the youth; even older people engage in sports regularly. This culture of exercise for all ages has spread in Oman as well, where you can see many seniors staying active, especially by the beach.”*
P18

Another social perception is that about gender, with women having more cultural restrictions when performing PA.


*“It is difficult for women to exercise with men in the same place.”*
P3

##### Weight Bias

Another social perception is about those who are overweight. One of the participants described how being overweight was an obstacle to being physically active. Being obese has a certain stigma associated with it that might deter those who are obese from actually practicing any type of exercise to reduce weight.


*“Ashamed of being overweight and playing sports, because of the difficulty in obtaining appropriate sports clothes and the fear of society’s view of him.”*
P9

##### Lack of Proficiency

Understanding what PA is and how to perform it were discussed differently among participants. When some participants were asked what their definition of PA is, some defined it as “exercise or sport” they regularly performed, whether it was weekly or daily, to reach a goal. 


*“Continuing sports, whatever the sport, and it continues daily every three days, or every week”*
P15

Others defined it accurately as a practice of several activities or as part of the daily routine, something practiced regularly, or movement that is more than *“normal”* P1 that produced *“positive results”* P10. However, when they are asked about barriers to PA, they will talk about barriers specifically to performing exercise.

A few participants mentioned that a barrier to PA was that they lacked the knowledge and skills to use exercise equipment or perform certain types of exercise, which deterred them from learning about new and/or different types of PA.


*“Sometimes it hinders me a lot when I enter a new field, and I don’t know how to use it, …or I don’t know how to use some tools or equipment …then I have to go back to something I know I’m familiar with.”*
P5

##### Gym-Centric Belief 

A variety of participants perceived PA as exercise performed only at the gym; hence, they might be incapable of being physically active without accessing a gym. For others, they are not able to sustain their exercise routine without going to the gym, even if they have exercise equipment in their private homes.


*“One of the obstacles is that one must go to a gym to stay active.”*
P2

##### Injury

Injury plays an important role in being incapable of being physically active. Not having the correct knowledge or the skills to perform certain types of PA can lead to bad outcomes.


*“Because sometimes, not knowing the correct technique for certain exercises can lead to injuries or even cause one to stop engaging in this sport.”*
P1

### 3.3. Part 2: Campaign Reflections

#### 3.3.1. Engagement

Most participants suggested ways of increasing engagement with social media awareness campaigns through the use of tools to increase exposure and maintain interest, such as including the use of incentives and/or rewards, the use of better marketing through mass media, and the launch of a specific social media account for the campaign by itself, so it does not get lost in other GHC account content. Lengthening a campaign’s time exposure and using traditional offline outlets such as malls were also discussed as a way to increase engagement with social media campaigns. 

One of the participants suggested that there is a need to employ a media spokesperson who must be qualified in the field and talented at convincing people to be more physically active. 

Engagement should be increased through focusing on the content, such as by using representatives of the targeted population in visual campaign content so they can be relatable, using true stories to emphasize the reality of it, basing content on scientific evidence, and explaining the health benefits of PA.


*“Benefit from scientific studies and research, practical ideas presented by the campaign.”*
P6


*“We are used to seeing athletes, and we are used to seeing sports people recommending sports, and we are used to seeing healthy people… but the thing we are not used to is seeing people who don’t do sports.”*
P7


*“Using persuasion with realistic stories from the same community and clarifying the benefits of sport and its impact on mental and physical health.”*
P9

Some participants suggested tailoring content to the target population by age, or available resources that participants already have, to increase engagement and ensure the sustainability of the campaign’s effect.


*“If it were, for instance, a real video with real people, it might attract adults but not necessarily those younger than them. However, videos with animation are more intriguing and exciting, and the graphics are better. Such videos draw people in. So, apps and videos that have this new animation are appealing.”*
P3

Interactive and entertaining content or tools are another engagement factor that some participants recommended, especially with educational content.


*“I suggest the campaign should have educational videos, and good interactive programs.”*
P3


*“By God, look, sister, in such situations, I believe that taking things too seriously can push people away, you know? The more entertaining the subject, the more it will attract and yield better results. I think the intended entertainment value should be included with it.”*
P4

#### 3.3.2. Acceptability

Acceptability of social media campaign platforms and content is important, and this point can be increased by diversifying campaign’s e-platforms while considering their usage and perception in the region. The latter point was emphasized by some participants. For example, when it comes to Twitter, this platform is associated mainly with political discussions. 


*“Because it contains many undesirable political contents. For instance, things that I personally don’t want. We have nothing to do with politics; many things don’t appeal to me. I access Twitter for specific purposes.”*
P3

In addition, they suggested using easier language to facilitate understanding of the campaign’s content. 


*“Information is better in the colloquial language that is close to everyone, not just intellectuals.”*
P9

Most responders said that the campaign should focus on the topic of PA as part of a lifestyle.

#### 3.3.3. Reach

Some participants suggested ways to improve the reach of future campaigns by aiming at different demographics, such as the elderly and younger people. A few participants suggested increasing reach through the use of already-installed cell phone apps and WhatsApp groups in a way that ensures sustainability. 


*“Using an application on iPhone and in other applications, for example, health, or calculating heartbeats, running and walking.”*
P15

A few participants mentioned the use of influencers in extending the reach of the campaign, including social media influencers who specialize in health content, others who have a huge number of followers and might be interested, or even famous, successful people who can be role models to others. 

#### 3.3.4. Design and Presentation

Participants who have seen the campaign’s content agreed that the information presented is easy to understand and touched upon different interesting information. The hashtag helped users to easily access the content on Twitter. The visual and audio campaign content, such as the infographics and the main video, were attractive and original.


*“From the beginning, the video was engaging, not just a boring awareness video. It posed questions that intrigued me. People were doing daily routines, and these people were like me—my friend, my cousin—discussing things in my community that I can incorporate into my daily life. It’s captivating and entertaining.”*
P8

#### 3.3.5. Perceived Outcome

Some participants mentioned the direct effect of the campaign on them, which was mainly educational and provided them with new and helpful information or corrected certain misconceptions. For some of them, it helped nudge them to perform PA or try new, different types of PA.


*“After the campaign, I have added different types of PA and maintained it daily.”*
P1

## 4. Discussion

This study aimed to explore barriers and facilitators to PA among GCC community members and their perceptions of social media awareness campaigns. Although awareness of PA and its benefits was present, only those who are/were physically active were motivated. Some lacked the capability, with certain social and physical opportunities acting as barriers. Indeed, motivation was a recurring domain as a facilitator, opportunity was a barrier, and capability was consistent for both facilitators and barriers.

Participants mentioned that PA is not always a priority. Studies have shown competing responsibilities and the low value placed on PA as reasons for this perception (15). Another barrier that might be linked to the value of PA is the reported lack of time. A review of previous evidence suggested that lack of time is not a significant barrier to be physically active, although it was commonly reported in studies of PA barriers. This might be due to participants using it as a justification for the lower value of PA rather than not having time [16]. PA locations, whether outdoor (e.g., park) or indoor (e.g., gym) have many barriers that deter people from being physically active, which is a common problem discussed in numerous GCC studies (4, 9, 10, 16). The belief that exercise can only be performed outside the home in a fitness center or outdoors in a park, which was a belief identified in the interview, adds to this barrier by making these locations the only place for PA as perceived by GCC community, rather than home gyms, for example, being suitable locations. 

There is a perception that being elderly, female, or overweight means that being physically active is not easy, despite the fact that these groups are at risk of various chronic diseases [26]. This perception is supported by the reported low levels of PA among these groups in systematic reviews of PA in the region [8,10,16]. 

The lack of PA proficiency seemed to steam mostly from unawareness of PA’s definition. That could hinder following or understanding PA-promoting messages [27]. 

The perception of the health value of PA was a major incentive for participants. Having previous experience, especially at a younger age, enjoying it, and having the skills to manage time between PA and life, as well as the use of behavior change techniques, were enablers. Indeed, a study conducted in Saudi Arabia showed that half of their sample practiced PA because of their awareness of its health value [28]. A systematic review of interventions for PA in GCC recommended the use of behavior change techniques (BCTs), such as monitoring behavior, as one effective intervention to promote it [29]. 

This study reiterates other previous studies about the unavailability of opportunities for the GCC community to practice PA [8,13,16,29]. Indeed, the results emphasize an ecological and multilevel view for intervention opportunities [30], such as the lack of suitable indoor exercise facilities and the unsuitable climate for outdoor PA in the region. As one 2-year cohort study in the region that looked at the effect of temperature on the number of steps, it showed that increased temperature and humidity were associated with a reduction in the number of steps taken [31]. These results match the findings in a systematic review of climate change and PA [32]. 

This study showed the acceptance of social media awareness campaigns as a potential channel for delivering health awareness messages. In a review that looked at engagement and acceptability of social media campaigns amongst vulnerable families (e.g., low-income parents of younger children), they found it a suitable and highly acceptable channel [33].

Participants recommended different ways to increase engagement, acceptability, and reach with future campaigns. A systematic review of reviews that looked at mass media PA campaigns that promoted it as a social norm were shown to be effective in changing behavior. Campaigns with longer durations, greater intensities, that were tailored to the target audience based on characteristics such as gender and age, and that were multi-component were more effective [12]. This aligns with WHO’s recommendation for making PA campaigns part of a multilevel, multicomponent, sustainable intervention to be effective in raising levels of PA [2].

Participants mentioned that the campaigns’ main perceived outcome was that they raised awareness about PA. This observation aligns with findings that PA campaigns can boost awareness and foster positive attitudes towards PA, but their effects on actually increasing levels of PA were mixed [12].

Qualitative research offered a deeper understanding of the cultural, social, and personal dynamics influencing PA in the GCC. This method provided insights into how individuals perceive PA within their cultural and environmental context, offering crucial information for designing relevant health interventions and emphasizing the results of previous quantitative studies [13,16,29,34]. Awareness campaigns, informed by this study, can be more effectively tailored to address the specific barriers and motivators to PA identified in the GCC. Messages can be developed using these findings. The interplay between qualitative research, the barriers and facilitators of PA, and the strategic execution of awareness campaigns creates a robust framework for addressing public health challenges in the GCC. This integrated approach ensures that interventions are not only evidence-based but also culturally congruent and socially engaging, thereby maximizing their impact on the health and well-being of the GCC community.

An in-depth exploration of each GCC country should be conducted to understand their perceptions, as this study hints at differences in perceptions about barriers, such as advanced age. A study in Oman showed that older age was not a barrier [35] to being physically active, which aligns with the perceptions of Omani participants but not other participants. It would also allow us to understand if the change is related to the interventions implemented in the region. Research and public policies need to be developed to address these barriers and facilitate PA in GCC communities. Opportunities to increase PA in the environment should be implemented, and awareness should be increased about home-based PA and the importance of being physically active even without going to a fitness center or an outdoor location to exercise. This should be reinforced with messages about reducing physical inactivity and differentiating between it and PA [27]. The health value of PA should be enhanced for all genders and ages but with messages tailored to the interests and needs of each group. 

### Strengths and Challenges

This is the first qualitative study, to our knowledge, to explore perceptions of barriers and facilitators to PA and PA awareness campaigns from the different GCC communities, which encompassed different genders, ages, and employment statuses. It gave a context to previous quantitative studies about the barriers to PA in the region [8,10,13,16,29]. The study also used a behavior change model to collect and analyze the data, providing a structured and comprehensive view of barriers and facilitators to PA. Two coders analyzed all the transcripts independently, improving rigor and interpretation. 

The study had some limitations, such as the possibility of social desirability bias; however, the interviews were virtual, obtaining the participants through online accounts and emails, and having interviewers from an independent organization means that this might be reduced [36,37,38]. Recruiting participants online might have excluded those who do not have access to social media to obtain their perspective about PA, which widens the digital divide that was introduced with an online-based campaign. 

## 5. Conclusions

Understanding the interplay of facilitators and barriers to PA within the GCC community provides a foundation for developing effective PA campaigns. This research highlights the complexities of PA participation, emphasizing the role of values, perceptions, persistence, past experiences, social connections, and effective strategies while addressing competing priorities, environmental constraints, societal biases, and knowledge gaps. Post-campaign perspectives emphasize the need for relatable, engaging content and diversified outreach methods, offering insights for more effective interventions to encourage PA and healthier living in the GCC region. Research and public policies need to be developed to address barriers and facilitators to PA in GCC communities.

## Figures and Tables

**Table 1 behavsci-14-00174-t001:** Participants’ backgrounds.

Participant	Age (yrs)	Sex	Country	Occupation
1	43	Male	Oman	employed
2	58	Male	Qatar	employed
3	40	Male	Oman	employed
4	25–34	Male	Saudi Arabia	employed
5	32	Male	Saudi Arabia	employed
6	39	Male	Oman	employed
7	28	Male	Oman	employed
8	39	Female	Saudi Arabia	employed
9	40	Male	Bahrain	employed
10	27	Male	Saudi Arabia	employed
11	40	Male	Saudi Arabia	employed
12	22	Female	Saudi Arabia	Student
13	36	Female	Oman	employed
14	57	Male	United Arab Emirates	employed
15	29	Male	Kuwait	employed
16	45	Male	Oman	employed
17	20	Male	Saudi Arabia	Student
18	34	Male	Oman	Unemployed
19	40	Male	Saudi Arabia	employed

**Table 2 behavsci-14-00174-t002:** Summary of themes and subthemes.

Themes	Subthemes	
Part 1: Physical Activity		
COM-B Facilitators	Health value of PA Self-efficacy Persistence and variety Familiar consequences	Motivation
	Social support	Opportunity
	Behavior change techniques Time management Starting at a young age Enjoyment	Capability
COM-B Barriers	Not a priority	Motivation
	Outdoor restrictions Limited amenities Age and gender bias Weight bias	Opportunity
	Gym-centric belief Lack of proficiency Injury	Capability
Part 2: Campaign		
Campaign Reflections	Engagement	
	Acceptability	
	Reach	
	Design and presentation	
	Perceived outcome	

## Data Availability

The datasets used and/or analyzed during the current study are available from the corresponding author on reasonable request.

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
