# Peer review of "Perceptions of Arabian Gulf Residents and Citizens about Physical Activity and Social Media Awareness Campaigns: A Qualitative Study"

_behavsci, 2024, doi:10.3390/bs14030174_

Round 1
Reviewer 1 Report
Comments and Suggestions for Authors
You have done a great job in presenting a new and interesting research. i commend you for the translation work, from arabic to english. this is a very technical research using state of the art computer, soft wear, etc. The method overall is very good and you have shown a detailed insight into the physical activity of 18 people. some recommendations for you - describe number 96 and 97 give an example of how you did this analysis. The main problem with this paper is there is no true literature review. I recommend a literature review on all of the main topics - qualitative research, importance of physical activity, and the impact of being involved with a campaign. You should then compare what you found with this. The discussion is really the results of the study, whereas the discussion should be how your study fits in with the literature review. I suggest to add implications to these countries as a result of this study. This could also be divided into two different papers, one on the campaign and another on the physical activity. Good job just add some current or important literature.
Comments on the Quality of English Language
You did a very good job with english language.
Author Response
You have done a great job in presenting a new and interesting research. i commend you for the translation work, from arabic to english. this is a very technical research using state of the art computer, soft wear, etc. The method overall is very good and you have shown a detailed insight into the physical activity of 18 people. some recommendations for you:
Response: Thank you very much.
- describe number 96 and 97 give an example of how you did this analysis.
Response: Thank you. This can be found in detail in the context (lines 115-120) and data analysis (lines 158 to 179) sections.
- The main problem with this paper is there is no true literature review. I recommend a literature review on all of the main topics - qualitative research, importance of physical activity, and the impact of being involved with a campaign. You should then compare what you found with this. The discussion is really the results of the study, whereas the discussion should be how your study fits in with the literature review. I suggest to add implications to these countries as a result of this study.
Response: Thank you. This has been added in introduction (lines 35-39,56-58,60-70,77-84) and discussion (lines 527-538).
- This could also be divided into two different papers, one on the campaign and another on the physical activity.
Response: We thought about this but ultimately agreed to leave it as it is to enrich the study.
- Good job just add some current or important literature.
Response: Thank you. Added.
Reviewer 2 Report
Comments and Suggestions for Authors
I commend the authors for their venture into this specialized area of study. This work not only demonstrates scholarly rigor but also strikes a fine balance between theoretical comprehension and practical application. Kudos to the authors for crafting a piece that is both insightful and a testament to the value of dedicated research.
Although the title clearly indicates the subject of study, the main focus of the study appears to center around social media campaigns. In this regard, it may be beneficial to reflect this aspect in the title.
I suggest that the introductory section could provide a more comprehensive context of the published literature on social media campaigns and their relationship with physical activity. This information could better situate the reader regarding the state of the art in this domain, particularly concerning qualitative studies.
The authors, citing WHO (2017), note that awareness campaigns can reduce levels of physical inactivity. However, recent WHO recommendations also highlight the importance of reducing sedentary behavior (https://www.who.int/publications/i/item/9789240015128). Therefore, it is recommended to include some studies on sedentary behavior in the context of the Gulf Cooperation Council. Both recommendations and public health strategies promoting physical activity and discouraging sedentarism and physical inactivity should be grounded on clear and universal definitions of these concepts to avoid any ambiguous or misinterpreted messages (Thivel D, Tremblay A, Genin PM, Panahi S, Rivière D, Duclos M. Physical Activity, Inactivity, and Sedentary Behaviors: Definitions and Implications in Occupational Health. Front Public Health. 2018;6:288. Published 2018 Oct 5. doi:10.3389/fpubh.2018.00288) (https://doi.org/10.1186/s12966-017-0525-8).
To better contextualize the reader, it would be interesting to include some information about the characteristics of the interviewers - reflexivity (bias, assumptions, reasons, and interests in this research topic).
The universe of the sample is not just a practical limit aiding the sampling process but also plays a significant role in analysis and interpretation. It would be useful to know if there were any specific inclusion or exclusion criteria for participant selection.
A theme captures something important about the data in relation to the research question and represents a level of patterned response or meaning in the dataset. An important aspect in terms of coding is what counts as a pattern/theme. Clarifying the prevalence of the themes would be beneficial.
The 'Discussion' section relates the research findings in theoretical terms, detailing the theoretical background and the results obtained. However, it could be useful to also integrate ecological models in the discussion of the results, particularly in lines 465-470 (https://doi.org/10.1146/annurev.publhealth.27.021405.102100). In this way, the results could be explored through models that holistically emphasize the importance of integrating different levels of variables in explaining physical activity behaviors.
In the list of references, reference number 25 shows its doi, while the other references do not.
Author Response
I commend the authors for their venture into this specialized area of study. This work not only demonstrates scholarly rigor but also strikes a fine balance between theoretical comprehension and practical application. Kudos to the authors for crafting a piece that is both insightful and a testament to the value of dedicated research.
Response: Thank you very much.
Although the title clearly indicates the subject of study, the main focus of the study appears to center around social media campaigns. In this regard, it may be beneficial to reflect this aspect in the title.
Response: Thank you. This has been changed.
I suggest that the introductory section could provide a more comprehensive context of the published literature on social media campaigns and their relationship with physical activity. This information could better situate the reader regarding the state of the art in this domain, particularly concerning qualitative studies.
Response: Thank you. This has been added in lines 56-58,60-70,77-84.
The authors, citing WHO (2017), note that awareness campaigns can reduce levels of physical inactivity. However, recent WHO recommendations also highlight the importance of reducing sedentary behavior (https://www.who.int/publications/i/item/9789240015128). Therefore, it is recommended to include some studies on sedentary behavior in the context of the Gulf Cooperation Council. Both recommendations and public health strategies promoting physical activity and discouraging sedentarism and physical inactivity should be grounded on clear and universal definitions of these concepts to avoid any ambiguous or misinterpreted messages (Thivel D, Tremblay A, Genin PM, Panahi S, Rivière D, Duclos M. Physical Activity, Inactivity, and Sedentary Behaviors: Definitions and Implications in Occupational Health. Front Public Health. 2018;6:288. Published 2018 Oct 5. doi:10.3389/fpubh.2018.00288) (https://doi.org/10.1186/s12966-017-0525-8).
Response: Thank you for this interesting reference. This has been added as a recommendation in lines 493-494 and 548 and in the introduction line 50.
To better contextualize the reader, it would be interesting to include some information about the characteristics of the interviewers - reflexivity (bias, assumptions, reasons, and interests in this research topic).
Response: Thank you. This has been edited and added in study procedures lines 137-139.
The universe of the sample is not just a practical limit aiding the sampling process but also plays a significant role in analysis and interpretation. It would be useful to know if there were any specific inclusion or exclusion criteria for participant selection.
Response: Thank you. This has been included in lines 140-142.
A theme captures something important about the data in relation to the research question and represents a level of patterned response or meaning in the dataset. An important aspect in terms of coding is what counts as a pattern/theme. Clarifying the prevalence of the themes would be beneficial.
Response: Thank you for your recommendation. But this is not that type of qualitative research, as numbers or prevalence doesn’t reflect the theme’s importance. However, we mentioned the most common theme domains in discussion lines 475-476.
The 'Discussion' section relates the research findings in theoretical terms, detailing the theoretical background and the results obtained. However, it could be useful to also integrate ecological models in the discussion of the results, particularly in lines 465-470 (https://doi.org/10.1146/annurev.publhealth.27.021405.102100). In this way, the results could be explored through models that holistically emphasize the importance of integrating different levels of variables in explaining physical activity behaviors.
Response: Thank you very much for the valuable reference, it has been added (lines 503-504).
In the list of references, reference number 25 shows its doi, while the other references do not.
Response: This has been updated.
Reviewer 3 Report
Comments and Suggestions for Authors
Abstract needs some grammatical work. Could be more concise, reads more as an intro than an abstract?
Line 26 starts with and
Line 37 cite
Line 41- one of whose?
Line 45- what are the obesity/overweight/ type 2 diabetes numbers?
Line 50-53, is this info from this study? where is this coming from?
Line 56 rewrite
59- needs period
Line 61- grammar with ;
Line 71- formatting and grammar
past or present for the campaign? stay consistent.
Line 85 grammar
Line 87- this is all one sentence? This needs to be rewritten
Research design needs to be expanded upon.
Procedures- who was recruited? who were the participants that this was sent to? what is GA SA RA?
Line 113 cite com-b?
Table 1- why is there an age range for subject 4?
Missing table line between 18 and 19
Were these all active individuals??
Were there multiple participants that supported each of these other than the ones quoted?
Make sure everything is past tense.
How did participants get in the study if they hadn't seen the campaigns content? Line 422
Line 476- a review of reviews?
The purpose of the study was to identity PA barriers and facilitators of the GCC community and explore perceptions of social media campaigns. Were they recruited from the campaign that was mentioned?
The information you are presenting seems helpful, though rather limited, but the organization, grammar and presentation is difficult to understand the main points being delivered. Clarifying the recruitment etc and more methods may help to clarify.
Comments on the Quality of English Language
As mentioned, there are some grammatical/language errors that need to be fixed that would likely help to improve overall clarity of the paper.
Author Response
Abstract needs some grammatical work. Could be more concise, reads more as an intro than an abstract?
Response:Thank you. Edited.
Line 26 starts with and
Response:Thank you. Edited.
Line 37 cite
Response:Thank you. Edited.
Line 41- one of whose?
Response:Thank you. Edited.
Line 45- what are the obesity/overweight/ type 2 diabetes numbers?
Response:Thank you. Added (lines 44-46).
Line 50-53, is this info from this study? where is this coming from?
Response:Yes, from the referenced article. Added clarification.
Line 56 rewrite
Response:Thank you. Edited.
59- needs period
Response:Thank you. Edited.
Line 61- grammar with ;
Response:Thank you. Edited.
Line 71- formatting and grammar past or present for the campaign? stay consistent.
Response:Thank you. Edited.
Line 85 grammar
Response:Thank you. Edited.
Line 87- this is all one sentence? This needs to be rewritten
Response:Thank you. Edited.
Research design needs to be expanded upon. Procedures- who was recruited? who were the participants that this was sent to? what is GA SA RA?
Response:Thank you.This has been edited and these are the initials of authors. We replaced it with first names in the study procedures and data analysis sections.
Line 113 cite com-b?
Response:Thank you. Edited.
Table 1- why is there an age range for subject 4?
Response: The participant refused to give specific age and only used the range.
Missing table line between 18 and 19
Response:Thank you. Edited.
Were these all active individuals?? Were there multiple participants that supported each of these other than the ones quoted?
Response: Yes, there are but due to word limit we chose some of the quotes. You can differentiate between claims supported or individualized throughout the results .e.g. individualized: one participant…supported: many/some participants…
Make sure everything is past tense.
Response:Thank you. Edited.
How did participants get in the study if they hadn't seen the campaigns content? Line 422
Response: Apologies. Edited.
Line 476- a review of reviews?
Response: The referenced paper is a systematic review of reviews as labelled in its methodology:Stead M, Angus K, Langley T, Katikireddi SV, Hinds K, Hilton S, et al. Mass media to communicate public health messages in six health topic areas: a systematic review and other reviews of the evidence. Southampton (UK); 2019.
Edited to systematic review of reviews to avoid confusion.
The purpose of the study was to identity PA barriers and facilitators of the GCC community and explore perceptions of social media campaigns. Were they recruited from the campaign that was mentioned? The information you are presenting seems helpful, though rather limited, but the organization, grammar and presentation is difficult to understand the main points being delivered. Clarifying the recruitment etc and more methods may help to clarify.
Response:Thank you. Added (lines 140-142).
Round 2
Reviewer 3 Report
Comments and Suggestions for Authors
Great job addressing the comments and suggestions from both reviewers. The improvement in clarity helps the reader to understand the importance of the article.